# A Boolean network model of hypoxia, mechanosensing and TGF-β signaling captures the role of phenotypic plasticity and mutations in tumor metastasis

Grant Greene [ID], Ian Zonfa, Erzsébet Ravasz Regan [ID]*

Biochemistry and Molecular Biology, College of Wooster, Wooster, Ohio, United States of America

* eregan@wooster.edu

## Abstract

The tumor microenvironment aids cancer progression by promoting several cancer hallmarks, independent of cancer-related mutations. Biophysical properties of this environment, such as the stiffness of the matrix cells adhere to and local cell density, impact proliferation, apoptosis, and the epithelial to mesenchymal transition (EMT). The latter is a rate-limiting step for invasion and metastasis, enhanced in hypoxic tumor environments but hindered by soft matrices and/or high cell densities. As these influences are often studied in isolation, the crosstalk between hypoxia, biomechanical signals, and the classic EMT driver TGF-β is not well mapped, limiting our ability to predict and anticipate cancer cell behaviors in changing tumor environments. To address this, we built a Boolean regulatory network model that integrates hypoxic signaling with a mechanosensitive model of EMT, which includes the EMT-promoting crosstalk of mitogens and biomechanical signals, cell cycle control, and apoptosis. Our model reproduces the requirement of Hif-1α for proliferation, the anti-proliferative effects of strong Hif-1α stabilization during hypoxia, hypoxic protection from anoikis, and hypoxia-driven mechanosensitive EMT. We offer experimentally testable predictions about the effect of VHL loss on cancer hallmarks, with or without secondary oncogene activation. Taken together, our model serves as a predictive framework to synthesize the signaling responses associated with tumor progression and metastasis in healthy vs. mutant cells. Our single-cell model is a key step towards more extensive regulatory network models that cover damage-response and senescence, integrating most cell-autonomous cancer hallmarks into a single model that can, in turn, control the behavior of in silico cells within a tissue model of epithelial homeostasis and carcinoma.

**Data availability statement:** Latest version of software to reproduce all modeling results is publicly available at: https://github.com/Ravasz-Regan-Group/dynmod Model and simulation files allowing readers to reproduce all our results with a few command line runs are included in this submission as S1-S3 Files.

**Funding:** o All funding for this study was provided by The College of Wooster: the Henry Luce III Fund for Distinguished Scholarship, the Faculty Development Funds and the Life Sciences Endowed Funds for research support to ER, the Henry J. Copeland Independent Study Fund for research travel support to IZ, Wooster's Biochemistry and Molecular Biology Program to GG for support to present his work. The funders had no role in study design, data collection and analysis, decision to publish, or preparation of the manuscript.

**Competing interests:** The authors have declared that no competing interests exist.

## Author summary

The cellular environment in and around a tumor can aid cancer progression by promoting several cancer hallmarks. This environment can affect growth and cell death, as well as a phenotype change that renders cells migratory and invasive: the epithelial to mesenchymal transition. Hypoxia (low oxygen availability) promotes this transition, while the attachment of cells to soft matrices or high cell density environments hinders it. These influences are often studied in isolation. As a result, their crosstalk is poorly understood. To address this, we have built a network model of cellular regulation that integrates a cell's responses to hypoxia, the biophysical environment, and growth signals to model cell division, death, and the epithelial-to-mesenchymal transition in environments cells encounter during metastatic tumor progression. Our model reproduces a wide range of experimental cell responses and offers experimentally testable predictions about the emergence of cancer hallmarks, driven by mutations that affect the hypoxic response. Our single-cell model is a key step towards more extensive cell-scale models that also include cell aging and damage response. These, in turn, can serve as building blocks of a larger tissue model of healthy vs. cancerous epithelia.

## Introduction

Tumor formation is marked by the development of mutations leading to dysregulated cell growth and death. Accumulation of cancer hallmarks is further aided by the tumor microenvironment (TME) – the biochemical composition, biophysical structure of the tissue, and non-cancer cells residing inside and around a tumor, all of which exert a major influence on the development and progression of cancer [1,2]. The TME is both variable and dynamic, with primary and secondary metastatic sites found to have differing environments that further vary by cancer origin [3–5]. Interestingly, the TME can contribute to the initial formation of tumors, rather than simply relying on stochastic genetic change as originally believed for carcinogenesis [6]. As cancer remains the second leading cause of death globally, understanding the unique conditions that promote metastasis contributing to 70–90% of cancer mortality is paramount to identifying therapeutics to both prevent and treat it [7].

A rate-limiting step for metastasis is epithelial to mesenchymal transition (EMT), a process where cells gain mesenchymal characteristics and lose their apical-basal polarity; aiding their migration, tissue invasion, and dissemination to secondary sites [8]. Yet, the reverse process (MET) is also crucial for tumor growth at secondary sites [9,10]. EMT and MET often do not require mutations to arise; rather they emerge due to the phenotypic plasticity of cancer cells responding to the TME. The stepwise process of EMT is controlled by two central double-negative feedback loops [11,12]. The first switch flips when an epithelial cell transitions to a hybrid E/M state, requiring the activation of SNAI1 by extracellular signals, accompanied by repression of the epithelial microRNA miR-34 [13]. In this hybrid E/M state, cells

lose their apical-basal polarity, become migratory, yet maintain adherens junctions with neighbors to facilitate collective migration. To complete full EMT, loss of the second epithelial microRNA, miR-200, must occur in conjunction with high expression of the EMT factor ZEB1 [14]. This leads to the loss of E-cadherin and all adherens junctions, prompting a full mesenchymal phenotype. While these two feedback loops are but a small part of the regulatory network driving EMT, their central role is illustrated by the fact that hybrid E/M states were first predicted by a computational model of this circuit [11,12], later confirmed *in vitro* [15–18] and *in vivo* [19–22]. A series of computational models of increasing complexity followed, exploring the role of phenotypic stability factors [23,24], the connection between hybrid E/M cells and cancer stemness [18], spatial patterning in tumors [25], and the role of biological noise in EMT [26]. Most relevant for this study are models that capture the crosstalk between multiple EMT-driving extracellular signals [27]. Among these, the influential large Boolean model by Steinway et al. integrated TGFβ, Wnt, Notch, Hedgehog, and serval growth signaling pathways, predicting and verifying several interventions capable of controlling EMT *in vitro* [28,29].

In addition to well-characterized transforming signals such as TGFβ secreted by nearby cells, EMT is influenced by two key aspects of the TME. First, the concentration of oxygen is known to decrease in response to tumor growth [30]. The increased size of solid tumors coupled with poor vascularization and vessel integrity lead to hypoxia – defined as an oxygen supply of <40 mmHg or <6% $O_2$ [31,32]. Tumor cells respond to hypoxic conditions with the secretion of Vascular Endothelial Growth Factor (VEGF) to induce angiogenesis and meet their oxygen demand [33,34]. Yet, blood vessel formation within tumors is highly dysregulated and ineffective at relieving hypoxia [32,35]. Acute, moderate hypoxia, in turn, promotes invasion and metastasis – an alternative pathway for tumor cells to resolve the oxygen demand by migrating to oxygen-rich tissue. Second, EMT is highly dependent on the biophysical environment. Namely, high cell density hinders EMT [36,37], while the increased matrix stuffiness characteristic of fibrotic tissue promotes it [38,39]. Both hypoxia and the biophysical environment were shown to contribute to immuno-, radio- and chemotherapy resistance, furthering cancer mortality [40–44]. However, the context-dependent crosstalk of hypoxia and the biophysical environment in driving EMT and MET are not well understood.

The hypoxic response occurs through two mechanisms: a canonical response mediated transcriptionally by Hypoxia Inducible Factor (HIF) proteins, and a non-canonical loss of oxygen-dependent degradation enzyme function, increasing the stability of target proteins [31,45–47]. HIF proteins are heterodimeric transcription factors comprised of an oxygen-sensitive Hif-α domain paired with a constitutively stable Hif-β domain. Regulation of the Hif-α subunit occurs through the oxygen-dependent hydroxylation of its proline residues, performed by prolyl hydroxylase domain (PHD) enzymes (Fig 1A) [48]. This in turn allows for recognition by E3 ubiquitin ligase Von Hippel Lindau protein (pVHL), leading to rapid Hif-1α degradation [49]. Additionally, the Factor Inhibiting HIF (FIH) hydroxylase enzyme uses $O_2$ as a substrate for asparagine hydroxylation, inhibiting HIF binding to transcription co-activator p300 and thus blocking its transcriptional activity [50]. Most higher eukaryotes have three oxygen-sensitive HIFα isoforms; Hif-1α, Hif-2α, and Hif-3α [51]. The first discovered and best understood subunit is Hif-1α, stabilized in response to acute hypoxic stress [52]. In contrast, Hif-2α is preferentially stabilized during long-term hypoxic reprogramming, offering temporal control over HIF regulation [53,54]. Finally, Hif-3α forms a negative feedback loop with Hif-1α to fine-tune long-term hypoxia, while maintaining its unique transcriptional activity [55,56]. In comparison to the ubiquitously expressed Hif-1α, Hif-2α and -3α are more temporal and cell-specific [51,57,58].

Since the discovery of Hif-1α, several hundred genes have been discovered to be differentially expressed under hypoxia as a result of HIF transcription, affecting a wide array of signaling pathways [59–61]. Critically, both mesenchymal factors SNAI1 and ZEB1 are transcriptional targets of Hif-1α [62,63]. Additionally, several other mesenchymal markers are shared targets Hif-1α and the classical EMT-inducing TGF-β pathway [64,65]. Independent of HIF transcriptional regulation, non-canonical hypoxia responses involving the loss of oxygen-dependent PHD hydroxylation and the accumulation of reactive oxygen species (ROS) through the electron transport chain were also shown to promote EMT. For example, PHD/pVHL has been implicated in repressing Transforming Growth Factor-β (TGF-β), contributing to increased TGF-β

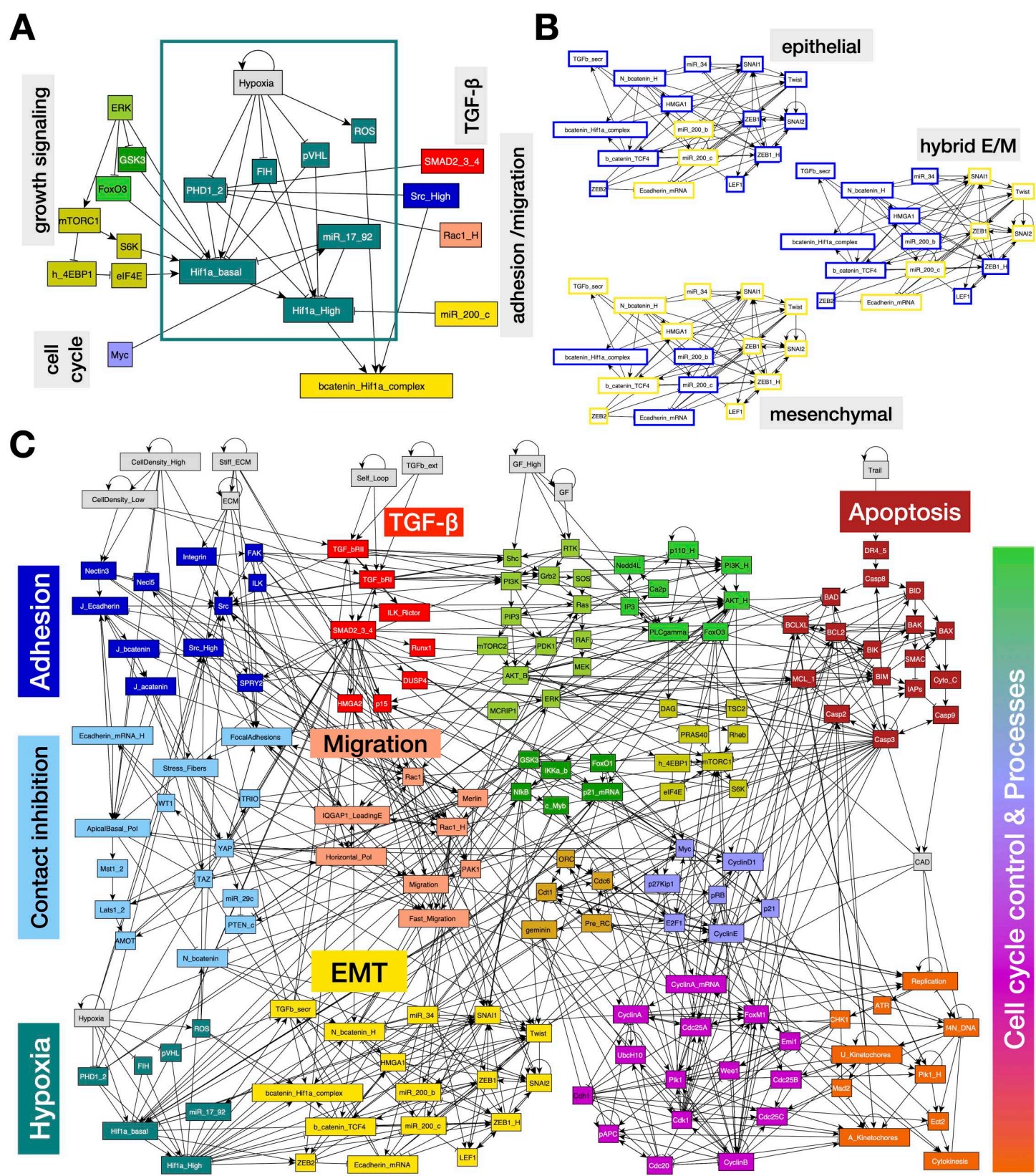

**Fig 1. The hypoxic response affects proliferative signaling and EMT.** A) Regulatory module depicting the canonical and non-canonical hypoxic response, along with influences from other signaling pathways and cell states. B) Three stable attractors of the isolated EMT regulatory switch (S2

File). *Orange/blue node borders:* ON/OFF. C) Modular network representation of our extended Boolean model. *Gray:* inputs representing environmental factors; *dark blue:* Adhesion signals; *red: TGFβ* signaling; *green*: Growth Signaling (*lime green*: basal AKT & MAPK, *bright green:* PI3K/AKT oscillations, *mustard*: mTORC1, *dark green: NF-κB, GSK3, FoxO1*); *dark red:* Apoptotic Switch; *light blue:* Contact Inhibition; *pink/light orange:* Migration; *light brown:* Origin of Replication Licensing; *lilac:* Restriction Switch; *purple:* Phase Switch; *dark orange:* cell cycle processes; *yellow:* EMT switch; *teal:* hypoxic response; →: activation; ⊣: inhibition.

expression under hypoxia [66]. Moreover, ROS accumulation caused by increased glycolysis under hypoxia contributes to Src activation and PHD inhibition, furthering invasion, metastasis, apoptosis resistance, and treatment resistance [67–69]. Src tyrosine kinase activity upregulates ERK [70] and is required for the loss of cadherin-dependent cell-cell contacts under hypoxia [71].

Despite the delineation of molecular pathways responsible for canonical and non-canonical hypoxic responses, downstream effects on the cell can appear paradoxical. For example, the Hif-1α-driven Warburg effect is required for proliferative metabolism [72], yet extreme hypoxia can inhibit proliferation [73,74]. While metabolic reprogramming under chronic hypoxia aids proliferation [75], acute stabilization of Hif-1α under hypoxia has been shown to prevent proliferation through strong Myc antagonization [74]. Similarly, while hypoxia induces apoptosis, hypoxia-driven EMT confers resistance to anoikis (apoptosis triggered by loss of adhesion) [76,77]. The pro- and anti-apoptotic effects of hypoxia are suggested to be distinguished by severity and length, governed by two competing cellular compartments: the cytosol vs. mitochondria. On one hand, hypoxia-driven loss of mitochondria membrane potential and decreased ATP production can increase mitochondrial membrane permeability, resulting in cytochrome C release and downstream caspase cleavage under severe hypoxia [78,79]. Under moderate hypoxia, localization of Hif-1α to the mitochondria prevents oxidative-stress-induced apoptosis by reducing mitochondrial transcription and ROS production [80]. On the other hand, Inhibitor of Apoptotic Protein-2 (IAP-2) expression increases under hypoxia, providing cytosolic prevention of apoptosis independently of Hif-1α activity [81]. Loss of HIF function attenuated these behaviors, demonstrating the necessity of the canonical pathway. These differing outcomes highlight the need to map the precise combinations of environmental conditions leading to each response. Furthermore, the effect of the biophysical environment on these diverging responses remains largely unexplored.

The tumor microenvironment is heterogeneous, both in terms of the spatial distribution of different tumor-associated cells and temporal differences in its molecular composition. Thus it is important to understand how the biophysical environment modulates the phenotypic diversity of tumor cell populations [82,83]. For example, single-cell sequencing experiments have found heterogeneous expression of epithelial and mesenchymal markers emerging from the same tumor site, confirming a broad range of states within the EMT spectrum [22,84,85]. As oxygen availability in tumors is also not static, it is important to understand the interplay between these two highly dynamic pathways: the hypoxic response and the induction of EMT in response to the biophysical environment.

The goal of this study is to map the way hypoxia alters these responses in a biophysical milieu-dependent manner. In a previously published Boolean model of mechanosensitive EMT, we reproduced known effects of the biophysical environment on EMT and TGFβ signaling. We predicted that in the absence of TGFβ, the decision between no response, partial EMT, and full EMT in mitogen-stimulated epithelial cells on a stiff ECM is dictated by cell density [86]. Here we expanded this model by including key regulators of the canonical and non-canonical hypoxic response and integrated their effect on growth factor signaling, proliferation, EMT, and apoptosis. Our model reproduces the requirement of Hif-1α for normal proliferation [87], as well as the antiproliferative effects of strong Hif-1α stabilization during hypoxia [73,74]. Additionally, we capture the molecular mechanism of Hif-1α mediated SNAI1/2 and ZEB1 upregulation [62,63,88] and reproduce hypoxia-driven EMT [63,89]; independent of TGF-β(66) but dependent on a stiff ECM [90]. Our model also reproduces hypoxic protection from anoikis [76]. We offer experimentally testable predictions about the context-dependent effects of TGF-β signaling on hypoxia-mediated EMT, as well as VHL loss on cancer hallmarks with or without secondary oncogene

activation. Taken together, we offer a mechanistic, predictive framework to synthesize the signaling responses responsible for hypoxic behaviors associated with tumor progression and metastasis in healthy vs. mutant cells.

## Results

### Expression of Hif-1α is required for proliferation

To model the way hypoxia contributes to EMT and proliferation, we expanded our previous mechanosensitive Boolean network, which includes a detailed cell cycle control circuit driven by growth signaling (MAPK, PI3K/Akt, mTORC) integrated with cell-cell and cell-ECM adhesion, contact inhibition, control of EMT, and apoptosis [86]. Here we incorporated a hypoxia module with canonical and non-canonical response mechanisms [86] (Fig 1A; model in S1 File). This module captures the acute, moderate hypoxic response, focused on Hif-1α and its downstream transcriptional activity [91]. In cells, Hif-1α activation and signaling have two functionally distinct regimes, requiring two Boolean nodes to properly capture its effects. Moderate Hif-1α is activated in response to growth factor signaling under normal $O_2$ conditions to boost glycolysis and proliferative metabolism; we encode this as *Hif1a_basal* = ON. A stronger activation occurs in response to hypoxic conditions or TGF-β signaling, typically in combination with deactivation of Hif-1α inhibitors such as pVHL (modeled as *Hif1a_High* = ON). Next, we included Hif-1α mediated transcription of mesenchymal markers including SNAI1/2, ZEB1, LEF1, β-catenin (and its subsequent nuclear localization/ complex formation with Hif-1α) [46,62,63,92]. Fig 1B shows the three stable states (fixed-point attractors) of the isolated EMT module, matching the expected activity of EMT regulators in three distinct phenotypic cell states along the EMT spectrum; namely epithelial (top), hybrid E/M (middle), and mesenchymal (bottom). The full network is shown in Fig 1C, while the Boolean rules and detailed biological justification for each of the 165 nodes and 763 links are included in S1 Text.

First, we validated our model by comparing its behavior to 150 experimental assays curated from 13 primary literature papers focused on EMT and hypoxia signaling. The 150 assays covered cell behaviors in response to 19 distinct protein or microRNA perturbations (knockdown/ overexpression) and involved 229 unique statistical tests, listed in S1 Table. Our model reproduced 92% [138] of the 150 assays (207 of 229 statistical tests; auto-generated figures in S12 Fig). Of the 12 failed assays marked in S1 Table, 7 involved experiments that directly contradict others on the list; the others pointed to model limitations detailed in our *Discussion*. Below, we showcase key validation experiments related to hypoxic signaling, along with a series of model predictions.

To test whether the updated model can reproduce the context-dependent influence of Hif-1α activation on proliferation, we used synchronous Boolean update to map the model's attractors in each cellular environment (*Computational Methods IV*). As our starting cell state, we chose a quiescent epithelial phenotype under normoxia, low mitogen exposure sufficient for quiescence, plated at moderate density (e.g., at a monolayer's edge). We then exposed this cell to increasing concentrations of mitogen (Fig 2A, *full time-course:* S1 Fig). As Hif-1α is a transcriptional inducer of glycolytic genes, we expected a moderate Hif-1α activity increase despite the presence of oxygen, required for maintaining the energy demand associated with proliferation [93–95]. Indeed, our model showed intermittent, moderate Hif-1α activation preceding each division (Fig 2A, *arrows*). The average activity of the *Hif1a_basal* node in a population under increasing mitogen exposure went up in tandem with Myc, Cyclin D1, and Cyclin E (Fig 2B), parallelling the increase in cell cycle progression (Fig 2C). Indeed, the loss of Hif-1α under normoxia reduced proliferation in our model (Fig 2D), as demonstrated *in vitro* [87]. Yet, high levels of Hif-1α of accumulation were also associated with cell cycle arrest due to Myc inhibition [73,74], reproduced by our model by overexpression of the *Hif1a_High* node in strongly mitogen-stimulated cells (Fig 2E). Strong Hif-1α activation blocks the cell cycle via Myc repression, as demonstrated by partial cell cycle rescue in cells with hyper-active Myc (S2 Fig). In summary, both over and under-expression of Hif-1α severely limit proliferation in our model, as described in the experimental literature.

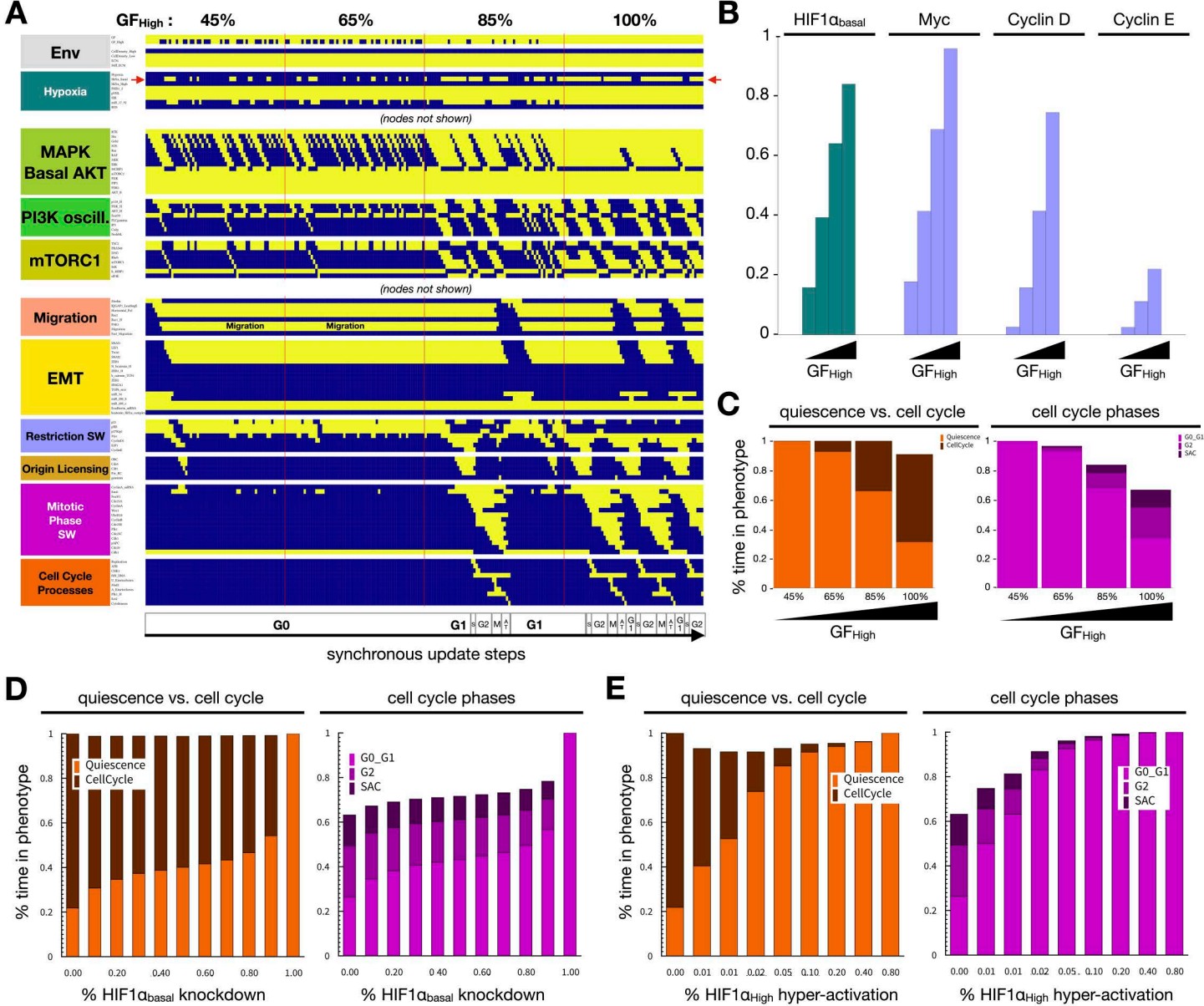

**Fig 2. Hif-1α is required for proliferation under normoxia.** A) Dynamics of relevant regulatory molecule expression/activity during exposure of a quiescent cell to increasing mitogen signaling (60 update-steps of 45%, 65%, 85%, and 100% high GF; full network dynamics in S1A Fig). *X-axis:* update steps annotated by cell cycle phase (G0: quiescence; G1: start of cell cycle entry; S: DNA synthesis; G2: growth phase 2; M: metaphase; A T: anaphase, telophase, cytokinesis); *y-axis:* nodes organized by regulatory module; *yellow/blue:* ON/OFF; *red arrow:* Hif1a_basal node. B) Average expression of *Hif1a_basal* (*teal*), Myc, CyclinD1, and CyclinE (*lilac*) in the four consecutive time windows with increasing GF_High input (45%, 65%, 85%, 100%) in an ensemble of 10,000 simulations representing individual cells. C) Fraction of time cells spend in quiescent (*orange*) vs. proliferative (*dark red*) states (*left*) and fraction of time a cell's state matched the G0_G1, G2, or Spindle Assembly Checkpoint (SAC) state of the phase switch (*right*) in subsequent time windows with increasing growth factor (45%, 65%, 85%, 100%) in an ensemble of 10,000 cells. D) Fraction of time cells spend quiescent (*orange*) vs. proliferative (*dark red*) states (*left*) and fraction of time cell states match the G0_G1, G2, or SAC state of the phase switch (*right*) at increasing levels of *Hif1a_basal* knockdown (*autocrine TGF-β:* 5% TGFb_secr knockdown). E) Fraction of time cells spend in quiescence (*orange*) vs. in cell cycle (*dark red*, *left*) and fraction of time cell states match the G0_G1, G2 or SAC state of the phase switch (*right*) at increasing levels of *Hif1a_High* hyper-activation (*log2 scale:* 0, 0.625, 1.25, 2.5, 5, 10, 20, 40 and 80% Hif1a_High=ON; *autocrine TGF-β:* 5% TGFb_secr knockdown). *Length of time-window for continuous runs:* 500 steps (~24 wild-type cell cycle lengths); *total sampled live cell time:* 100,000 steps; *update:* synchronous; *initial condition for all sampling runs:* epithelial cells in GF:1, CellDensity_High:1, Stiff_ECM:1, Trail:0, Self_Loop:1, TGFb_ext:0, Hypoxia:0; *environment of sampling runs:* saturating growth signals (GF_High:1), moderate density (CellDensity_Low:1).

## Hypoxia arrests proliferation and induces EMT

Having demonstrated the role of Hif-1α expression in proliferation under normoxia, we next assessed the effect of the hypoxic response on proliferation and EMT. As described above, Hif-1α overexpression in the presence of oxygen leads to cell cycle arrest due to Myc repression (S2 Fig) [74]. Hypoxia induced a similar cell cycle arrest in our model, as lack of oxygen blocked PHD2 hydroxylation and downstream VHL-mediated Hif-1α degradation [92] (Fig 3A). In addition to cell cycle arrest, our simulation showed hypoxia-induced EMT in cycling cells (Fig 3A), as expected from experimental literature [87,90,96]. Namely, hypoxia resulted in Hif-1α mediated transcription of mesenchymal transcription factors SNAI1/2, ZEB1, LEF1, and nuclear β-catenin, generating a mesenchymal phenotype and complete loss of epithelial markers (S3A Fig) [63,92,97]. In addition, non-canonical (Hif-1α independent) stabilization of IKKα due to a lack of oxygen-dependent hydroxylation by PHD led to NF-κB activation (S3A Fig) [98].

As seen in Fig 3A, hypoxia triggers autocrine TGF-β signaling due to a combination of oxygen-dependent loss of VHL activity [66], Hif-1α mediated activation of TGF-β converting enzyme furin [99], and critically, secretion of TGF-β induced by EMT transcription factors [46,100]. Yet, this effect is a consequence rather than a driver of EMT under hypoxia, as TGF-β receptor I knockdown does not block hypoxia-induced EMT – though it slows it in mild hypoxia (Fig 3B-3C). That said, lack of TGF-β signaling does weaken cell cycle arrest in the presence of mitogens (Fig 3D). Moreover, hypoxic EMT in VHL independent (Fig 3B-3C), as PHD2-mediated hydroxylation and degradation of Hif-1α requires oxygen even in VHL-overexpressing cells (S3B Fig) [48,101]. In contrast, TGF-β can only stabilize Hif-1α in the absence of VHL (S3C Fig) [101]. As EMT transcription factor induction occurs independently through either Hif-1α or SMAD2/3/4, the two pathways do not rely on each other to promote EMT (S3D Fig).

## The biophysical environment and oxygen availability modulate EMT and its reversal

Experimental literature on the crosstalk between the hypoxic response and ECM primarily focuses on hypoxia-induced remodeling by fibroblasts, or induction of the Warburg effect [102–106], leaving the interplay of hypoxia and ECM stiffness or cell density underexplored. One of the few direct studies on the dependence of EMT on ECM stiffness under hypoxia demonstrated that mesenchymal marker induction under hypoxia positively correlates with matrix stiffness in vitro[12]. To test whether our model can reproduce this, we started our next simulation with a quiescent epithelial cell on a soft ECM, at low density and under hypoxia (arrested due to an inability to stretch, as well as hypoxia). We then gradually increased ECM stiffness by stochastically turning ON the *Stiff_ECM* node, until its saturation (*Stiff_ECM=ON*). As shown in Fig 4A, mesenchymal markers gradually turned on starting from *Stiff_ECM=0.5*, and cells maintained a fully mesenchymal state by *Stiff_ECM=0.75* (complete time-course: S4A Fig). In contrast, cells under normoxia underwent partial EMT and entered the cell cycle, never reaching high levels of ZEB1 to repress miR-200c and E-cadherin expression (Figs 4B and S4B). The average SNAI1 activity of a heterogeneous population of *in silico* cells at increasing ECM stiffness closely resembled its relative expression *in vitro* (S5A Fig). Additionally, the effect of matrix stiffness on migration was stronger under hypoxia, aligning with the study's wound-healing results (Fig 4B, *middle row*). Lastly, our model predicts that matrix stiffness affects hypoxia-induced EMT more profoundly than TGF-β induced EMT (S5B Fig). The EMT-induction threshold of hypoxia increases rapidly on softer ECMs, while the effect on TGF-β-induced EMT is significantly milder. We further predict that on soft ECMs that cannot support stress fiber formation and migration, only a combination of strong hypoxia and near-saturating TGF-β can induce EMT (S5B Fig, *left*). In contrast, high cell density blocks EMT in response to either signal, as well as their combination, while isolated mitogen-stimulated cells undergo biomechanically induced EMT even in the absence of hypoxia or external TGF-β (S5C Fig) [86].

## Loss of the Hif-1α repressor VHL aids metastatic cell behaviors

The development of clear cell renal cell carcinoma (ccRCC) is marked by mutations to Von Hippel Lindau gene, resulting in reduced expression or inactive forms of VHL protein and aberrant Hif-1α expression under normoxia as well as hypoxia

PLOS Computational Biology

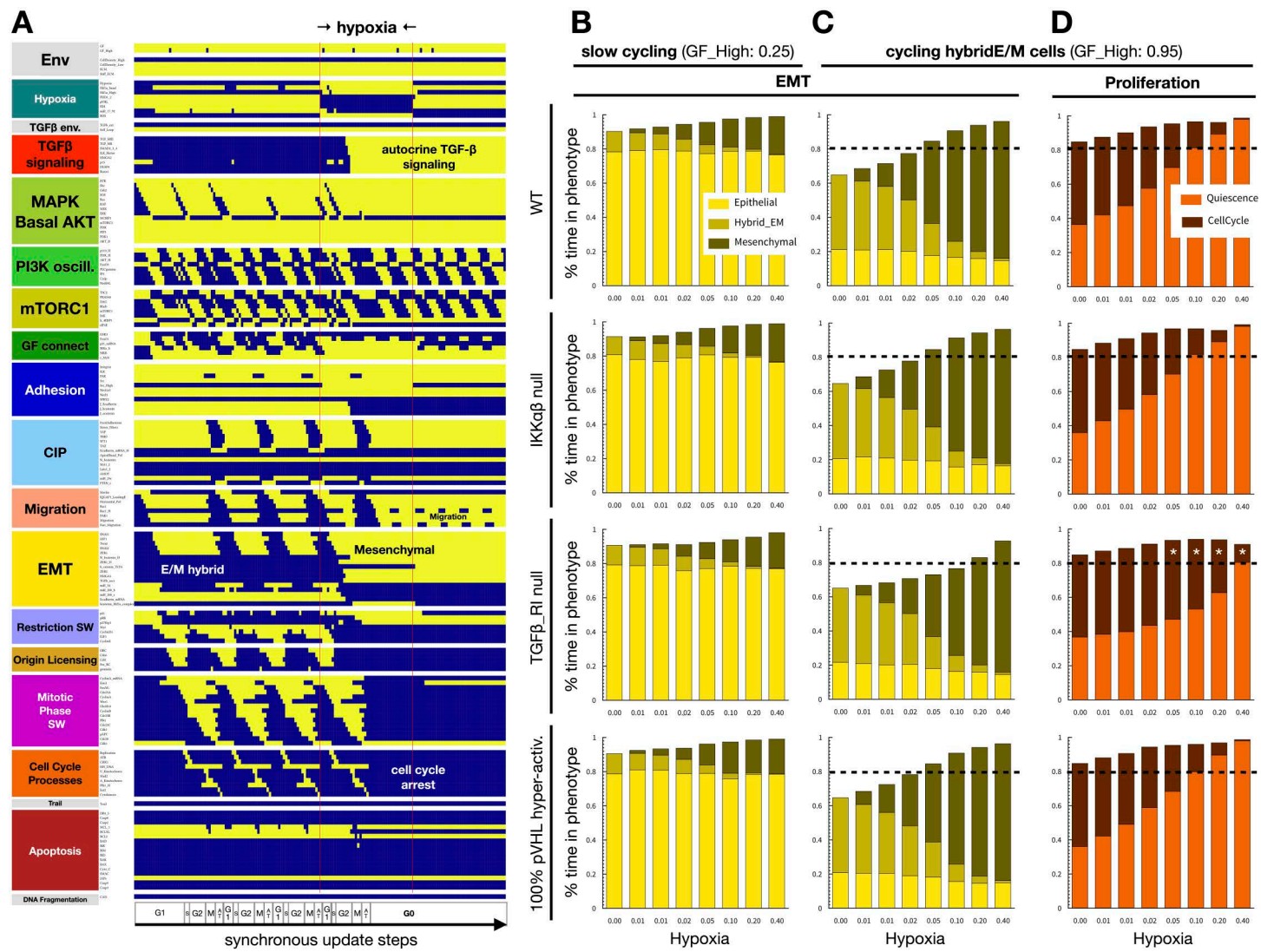

**Fig 3. Hypoxia induces EMT and blocks the cell cycle independently of TFG-β signaling.** A) Dynamics of regulatory molecule expression in a proliferating cell in response to hypoxia (80/ 40/ 40 update-steps of normoxia/hypoxia/normoxia at 95% saturating mitogen exposure and moderate cell density). *X-axis*: update steps annotated by cell cycle phase (G0: quiescence; G1: start of cell cycle entry; S: DNA synthesis; G2: growth phase 2; M: metaphase; A/T: anaphase, telophase, cytokinesis); *y-axis*: nodes organized by regulatory module; *yellow/blue*: ON/OFF; *black/white labels*: relevant phenotypes; *update*: synchronous. B-C) Fraction of time cells at moderate density exposed to weak (B) or strong mitogens (C) spend in epithelial (*yellow*), hybrid E/M (*dark yellow*), and mesenchymal (*mustard*) states at increasing hypoxia exposure. D) Fraction of time strong mitogen-exposed cells spend in quiescence (*orange*) vs. in cell cycle (*dark red*) at increasing hypoxia exposure. B-D) *1st row*: wild-type; *2nd row*: 100% IKKα/β knockdown; *3rd row*: 100% TGFβ-RI knockdown; *4th row:* 100% pVHL hyper-activation. *X axis* ($log_2$ scale): 0, 0.625, 01.25, 2.5, 5, 10, 20 and 40% hypoxia; *black dashed line*: 80% time; *white stars*: altered from wild-type; *length of time-window for continuous runs:* 100 steps (~5 wild-type cell cycle lengths); *total sampled live cell time:* 100,000 steps; *update*: synchronous; *initial condition for all sampling runs*: epithelial cells in GF:1, CellDensity_Low:1, Stiff_ECM:1, Trail:0, Self_Loop:1, TGFb_ext:0, Hypoxia:0; *environment of sampling runs*: GF_High = 0.25 (B) or GF_High = 0.95 (C) and moderate density (CellDensity_Low:1); *autocrine TGF-β loop:* 5% TGFb_secr knockdown.

[107]. This mutation contributes to ccRCC being the 13th most common malignancy throughout the world [108], as well as its reduced treatment efficacy and markedly high mortality rate [109]. While VHL loss is a hallmark of ccRCC, this mutation alone is unable to consistently induce tumorigenesis in mice [110,111]. Modeling VHL-deficient cells, we predict that the loss of VHL can drive metastatic cell behaviors under biophysical microenvironments that do not induce EMT in wild-type

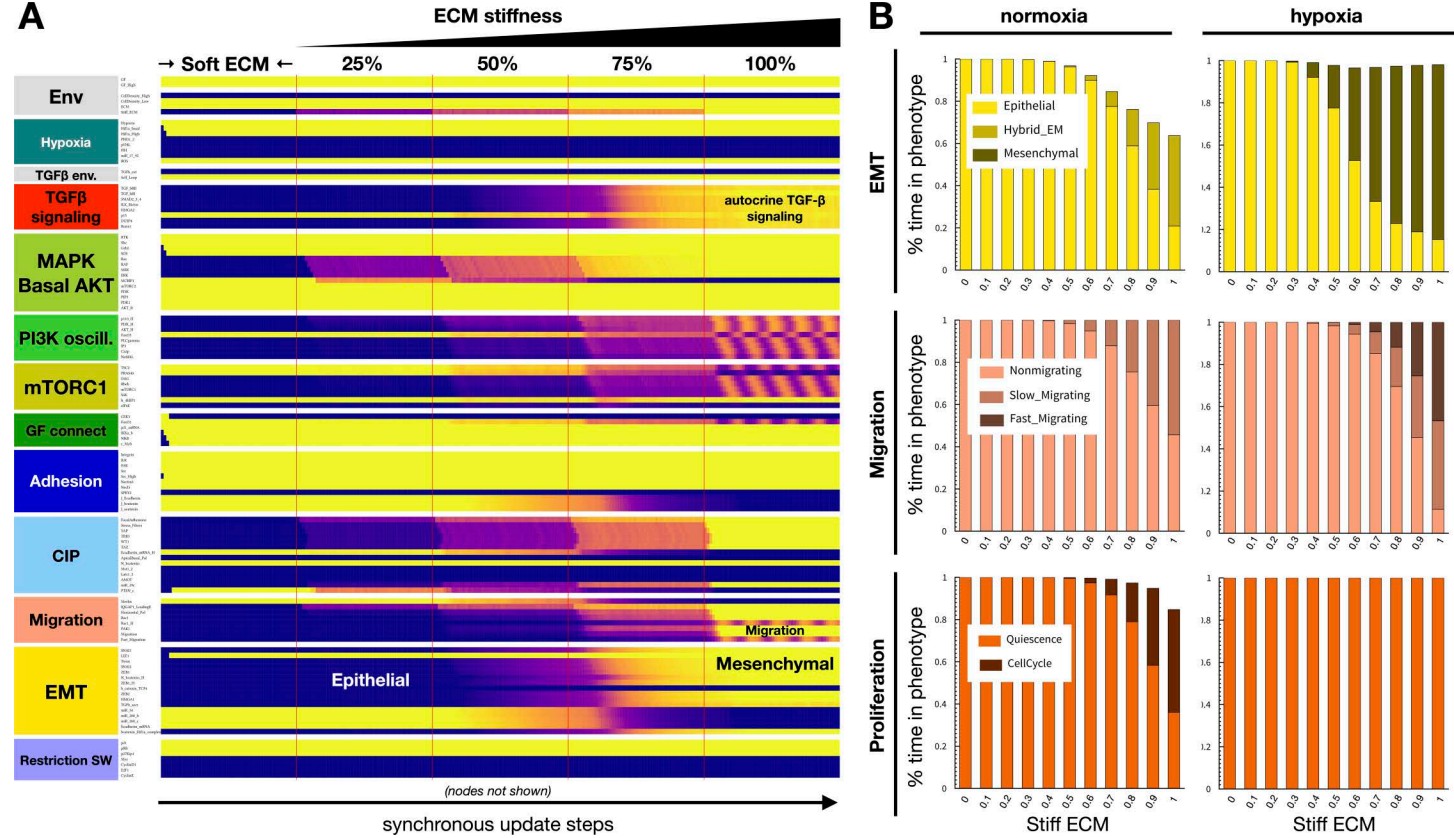

**Fig 4. A stiff ECM aids hypoxia-induced EMT.** A) Dynamics of relevant regulatory molecule expression in a quiescent cell on a soft ECM under hypoxia, in response to increasing ECM stiffness (50 update-steps of 0, 25, 50, 75, and 100% Stiff_ECM=ON at saturating mitogen exposure, moderate cell density, and 5% TGFb_secr knockdown). *X-axis:* update steps; *y-axis:* nodes organized by regulatory module; *yellow/purple/blue scale:* 100% ON/ 50% on/ 100% OFF; *black/white labels:* relevant phenotypes; *update:* synchronous; *full time-course:* S4A Fig. B) Fraction of time normoxic (*left*) vs. hypoxic (*right*) cells at moderate density and saturating mitogen exposure, plated on ECM of increasing stiffness, spend in: *top:* epithelial (*yellow*), hybrid E/M (*dark yellow*) and mesenchymal (*mustard*) states; *middle:* non-migrating (*light pink*), slow-migrating (*dark pink*) and fast-migrating (*brow-pink*) states; *bottom:* quiescence (*orange*) vs. in cell cycle (*dark red*). *Length of time-window for continuous runs:* 100 steps (~5 wild-type cell cycle lengths); *total sampled live cell time:* 100,000 steps; *update:* synchronous; *initial condition for all sampling runs:* epithelial cells in GF:1, CellDensity_Low:1, Stiff_ECM:1, Trail:0, Self_Loop:1, TGFb_ext:0, Hypoxia:0; *environment of sampling runs:* Hypoxia:0(*left*)/1(*right*), GF_High:1, CellDensity_Low:1; *autocrine TGF-β loop:* 95% (5% TGFb_secr knockdown).

cells. To this end, we simulated wild type vs. 90% VHL knockdown in cells on soft to stiff ECM, at densities ranging from isolated cells to dense fully confluent monolayers under normoxia (S6A Fig, *top left*). In the presence of a near-saturating growth stimulus, only cells with some access spreading space and a very stiff ECM underwent full EMT. In contrast, the combination of ECM and density values leading to EMT were significantly expanded in VHL-deficient cells. Loss of VHL boosted full EMT rates, particularly at moderate ECM stiffness and in cells that had only intermittent access to space to establish horizontal polarity (S6A Fig, *middle/bottom left*). While hypoxia blunted its effects, the boost to EMT was still significant at moderate hypoxia (S6A Fig, *middle/right columns*). As the loss of VHL stabilized Hif-1α, it also resulted in a full cell cycle arrest (S6B Fig), indicating that a boost to EMT in VHL-null cells would be counterbalanced by arrested growth.

Overall, our VHL-deficient simulations indicate that a combination of VHL loss and cell cycle-promoting oncogene activation is required for cancer progression, breaking Hif-1α and/or EMT-induced cell cycle arrest. Indeed, ccRCC tumors with a VHL missense mutation often harbor Cdk4/6 and/or Cyclin D amplification and are highly proliferative [112]. Moreover, the most proliferative subtype of ccRCC is both VHL deficient and has enhanced MYC activity [113]. To test whether

our model could reproduce proliferation rescue in VHL-null cells, we re-ran the above simulations at varying ECM stiffness and cell density in VHL-deficient cells with or without Myc and/or Cyclin D hyper-activation (S7A Fig). Intriguingly, only joint hyper-activation of Myc and Cyclin D could significantly boost the proliferation of VHL-deficient cells (S7B Fig, *top*). This co-occurred with a mild reduction in EMT compared to VHL deficiency alone (S7B Fig, *bottom*). In contrast, pairing Cyclin D with ccRCC-associated mutations upstream in the PI3K or MAPK pathway, such as PI3K/mTORC1/ERK hyper-activation or p21/PTEN loss [114], only led to weak cell cycle rescue (S7C Fig, *top*); though most of these combinations also mildly boosted EMT (S7C Fig, *bottom*). Pairing Myc with the same cancer mutations only yielded weak cell cycle rescue with p21 loss (S7D Fig).

## Hypoxia prevents anoikis and epithelial apoptosis in response to TGF-β

In addition to inducing a mesenchymal state, the lack of oxygen has been shown to block anoikis and apoptosis; further contributing to the mortality rate of hypoxic tumor formation [79,115]. As previously shown in Sullivan et al. [86], our mechanosensitive model can reproduce anoikis-mediated epithelial cell death in response to detachment from the extracellular matrix. To test if our model could also replicate hypoxia-mediated evasion of anoikis, we simulated detachment of a quiescent, isolated epithelial cell from a soft ECM exposed to saturating growth factors (Fig 5A; *full time-course:* S8 Fig). In contrast to normoxia, hypoxia repressed pro-apoptotic factor activation and conferred anoikis resistance [76,116]. This was due to the ability of cells suspended under hypoxia to maintain high levels of MAPK signaling and basal AKT activity. This was supported by hypoxia- and ROS-mediated Src upregulation, an effect that did not require integrin-mediated attachment and thus prevented anoikis (Fig 5A, *right*) [76]. The non-canonical hypoxic response also contributed to anoikis resistance through NF-κB nuclear translocation, reinforcing BCL-2 and BCL-X$_L$ expression [117,118]. Along similar lines, our model reproduces TGF-β induced apoptosis of epithelial cells on a soft matrix (Fig 5B, *left*) [119]. Specifically, the model depicts an increase in BAX/BIM signaling along with BCL-X$_L$ loss, followed by apoptosis. In comparison, cells exposed to TGF-β in a hypoxic environment are predicted to resist apoptosis through increased Src and MAPK activity (Fig 5B, *right*).

## Hypoxia and reoxygenation can drive the metastatic cascade in the absence of exogenous transforming signals

Next, we aimed to identify the biophysical microenvironments required for MET following reoxygenation, a crucial step in secondary tumor formation [120]. To simulate a metastatic cascade, allowing for both intravasation and extravasation from a solid tumor, we simulated a sequence of microenvironments designed to mimic the signals metastatic cells experience en route from a solid tumor toward a new metastatic site. To do this, we started with an epithelial cell at high cell density on a stiff ECM, such as the basement membrane [121] (Fig 6A, *pulse 1*). In a growing solid tumor, this cell is likely to experience hypoxia (Fig 6A, *pulse 2*). Yet, due to density-dependent contact inhibition of proliferation as well as migration, this epithelial cell is protected from hypoxia-driven EMT. For EMT to start, a decrease in local density must occur (Fig 6A, *pulse 3*). In a tumor setting this may be due to cell death in the neighborhood, potentially due to nearby anoxic areas of necrosis (not modeled) [79] or changes to constraining structures such as the basement membrane [122]. Migrating away from the solid tumor, a mesenchymal cell can now reach areas of lower density and adequate oxygen, while maintaining its mesenchymal state (Fig 6A, *pulse 4*).

As hypoxia induces self-sustaining autocrine TGF-β signaling [14], transitioning back to an epithelial state requires breaking the autocrine loop. This can happen due to non-saturating TGF-β secretion in transit through the bloodstream, where cells have no access to a stiff ECM to keep mechanosensitive EMT signals active (Fig 6A, *pulse 5*). As a result, upon entering the vasculature, detachment from the ECM and other cells is a crossroad with two phenotypic outcomes: cells that maintain a mesenchymal state avoid anoikis, whereas cells that revert to an epithelial state undergo apoptosis (Fig 6A, *pulse 5*). Surviving cells then transition to an epithelial state upon reaching high cell density in the new tissue (Fig 6A, *pulse 7*). From here, they can re-enter a proliferative hybrid E/M state once they escape contact inhibition [123]. This completes the metastatic cascade, leading to new tumor growth. In contrast to cells with non-saturating autocrine TGF-β

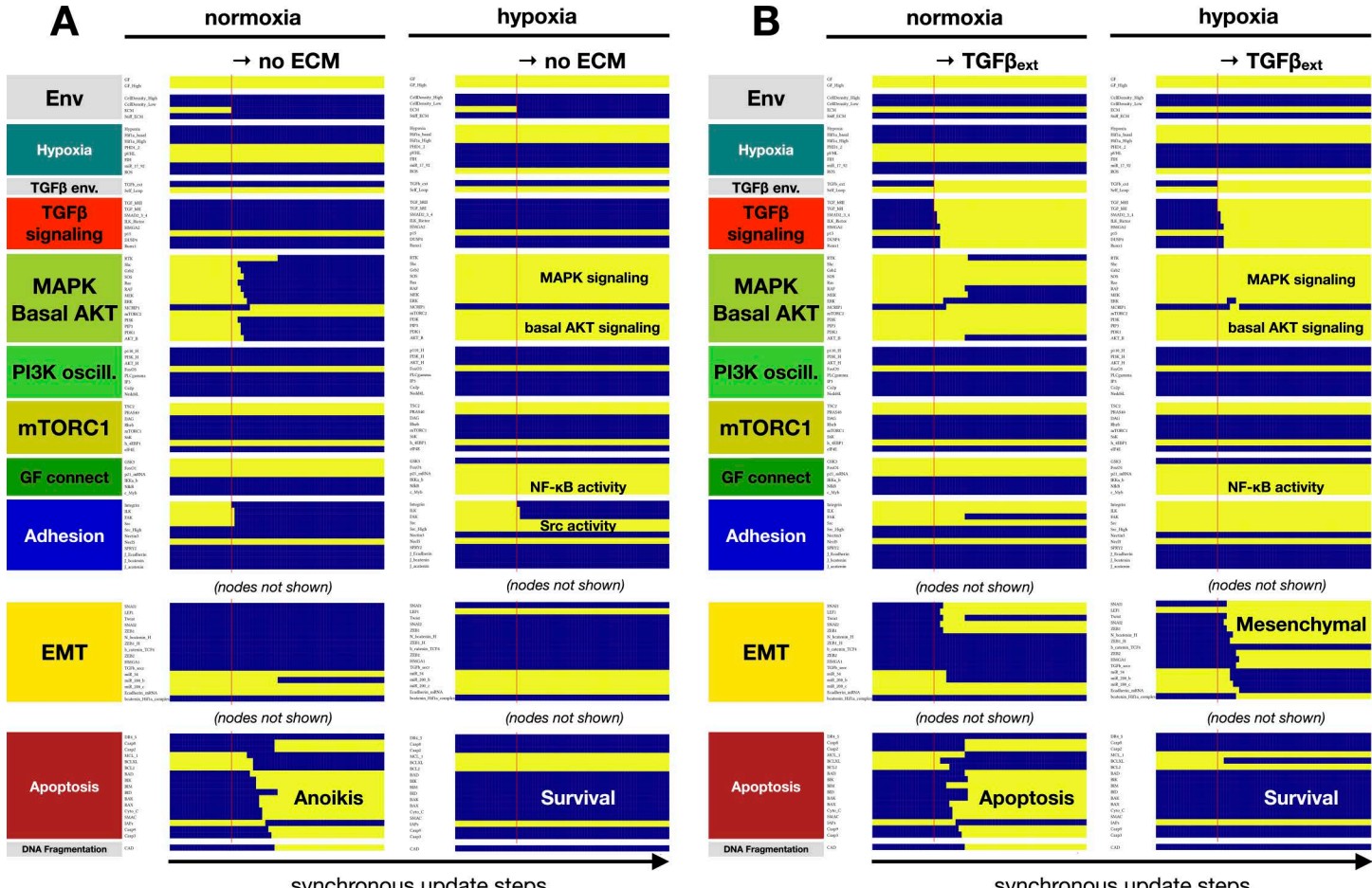

**Fig 5. Hypoxia confers anoikis resistance and protection from TGF-β induced epithelial cell apoptosis on a soft ECM.** A) Dynamics of relevant regulatory molecule expression in a quiescent cell detached from a soft ECM in normoxia (*left*) vs. hypoxia (*right*) (20 update-steps on soft ECM, 50 update-steps detached, 100% high GF). B) Dynamics of relevant regulatory molecule expression in a quiescent cell on a soft ECM, exposed to exogenous TGF-β in normoxia (*left*) vs. hypoxia (*right*) (20 update-steps no TGF-β, 50 update-steps 100% TGF-β). *X-axis:* update steps; *y-axis:* nodes organized by regulatory module; *yellow/blue:* ON/OFF; *black/white labels:* relevant phenotypes; *update:* synchronous; *full time-course:* S8 Fig.

signaling, saturating TGF-β secretion prevents apoptosis in nearly all cells undergoing the metastatic cascade (S9 Fig). That said, even slight perturbations of the autocrine loop, modeled by reducing *TGFb_secr*, are predicted to have a major impact on the rate of apoptosis in the bloodstream, as well as upon transitioning onto a soft ECM (S10 Fig).

Another way to visualize the environment changes driving EMT and MET is to highlight the cell state sequence the above steps push our model through, shown on a comprehensive map of all relevant stable model phenotypes (attractors) organized by microenvironment (S11 Fig). Taken together, our mechanosensitive model can reproduce hypoxia-driven EMT, and its reversal is based on a realistic sequence of its biophysical environments, depicting molecular signaling which occurs throughout the metastatic cascade.

## Discussion

Despite the prevalence of hypoxic tumor microenvironments during solid tumor progression, the combined effect of hypoxia and the biophysical environment on cancer hallmarks remains underexplored. Here, we have built a dynamic regulatory network model focused on depicting the molecular signaling that occurs in response to oxygen deprivation (Fig 1).

To do this, we incorporated a hypoxic response module into our previously published mechanosensitive EMT model [86]. Unlike published models exploring individual hypoxia-induced phenotypes [124–127], our model captures Hif-1α mediated transcription responsible for i) the induction as well as inhibition of proliferation (Fig 2), ii) propelling EMT in a mechano-sensitive manner (Figs 3 and 4), and iii) preventing cell death in response to a changing TME (Fig 5). Taken together, our model can reproduce cell state changes along a full metastatic cascade (Fig 6), providing insight into the effects of a continuously changing biophysical environment during secondary tumor formation. The resulting cell behaviors cover cell-autonomous aspects of several cancer hallmarks, including *sustaining proliferative signaling*, *evading growth suppressors*, *activating invasion and metastasis*, *resisting cell death*, and *unlocking phenotypic plasticity* [128].

1. *Sustaining proliferative signaling* & *evading growth suppressors.* Our model captures the paradoxical relationship between Hif-1α and proliferation. On one hand, Hif-1α is required for proliferation through its role in the upregulation of the glycolytic machinery, including GLUT1, LDH-A, ALDA, and more (modeled indirectly; Fig 2D) [129]. On the other hand, the increased stability of Hif-1α under hypoxic conditions is known to induce cell cycle arrest through functionally repressing Myc, a key driver of proliferation (Fig 2E) [73,74]. Indeed, our model predicts that Myc hyperactivation can rescue the cell cycle under hypoxia (S2 Fig). In comparison to the effects of *acute* oxygen deprivation captured by our model, *chronic* hypoxia increases proliferation through Hif-2α-induced Myc transcription, increased MAPK/ERK signal-ing, and YAP1 activation [130–132]. Capturing this temporal shift in our model is an area of future interest.

2. *Activating invasion and metastasis.* While the effects of hypoxia on EMT are well-documented, the mechanosensitive nature of this response, as well as its crosstalk with TGF-β signaling, have not been modeled individually or collec-tively. Our model shows that hypoxia can induce EMT in cycling cells (Fig 3) [87], as long as they have access to a stiff ECM and some space to spread (Figs 4 and S5) [90]. In contrast, normoxic cells only undergo hybrid E/M in all but the most EMT-promoting biophysical environments (S5 Fig). Once established, a mesenchymal state is main-tained regardless of hypoxia due to its ability to induce autocrine TGF-β signaling (Fig 3A), even in the presence of hyper-active VHL [101]. Yet, this autocrine signal is not required for EMT in hypoxic cells, as loss of TGF-β Receptor I does not block the transition (Fig 3B) [101]. In contrast, TGF-β can only boost Hif-1α stabilization in VHL-deficient cells. These nuances of crosstalk highlight the importance of modeling the biophysical environments and genetic interactions that compound the loss of VHL, a tumor suppressor pivotal to clear cell renal cell carcinoma (ccRCC) tumorigenesis [109]. These cancers have among the highest rates of metastasis and mortality [133]. Our model pre-dicts that VHL loss alone, known to predispose patients to cancer [134], can push cells at moderate density and/or on a softer ECM into full EMT. These cells undergo EMT instead of maintaining an epithelial state or a reversible hybrid E/M phenotype required for repairing epithelial damage, as expected in healthy cells (S6 Fig) [135]. Furthermore, we highlight the importance of secondary driver mutations that directly impact cell cycle entry, such as joint cyclin D and Myc amplification/hyperactivation [112,113]. These mutations overcome the cell cycle arrest mediated by hyperactive Hif-1α and/or autocrine TGF-β signaling (S7 Fig), helping cancer cells combine sustained proliferation with an invasive mesenchymal state.

3. *Resisting cell death.* Our model not only reproduces anoikis resistance of epithelial cells under hypoxia [76] but also predicts that hypoxia can protect epithelial cells on a soft ECM from TGF-β-induced apoptosis (Fig 5).

4. *Unlocking phenotypic plasticity.* Our simulation of the metastatic cascade, from hypoxia-induced EMT in a primary tumor through anoikis resistance in the bloodstream to cell density-induced MET at a secondary site, our model cap-tures the non-genetic phenotypic plasticity cancer cells tap into as they navigate different combinations of chemical and biomechanical environments (Fig 6). Interestingly, our model predicts that the ability to reach and sustain a fully mesen-chymal state, with no break in autocrine TGF-β signaling, serves as an environmental bottleneck for circulating tumor cells (S9 and S10 Figs). Moreover, intermittent (non-saturating) TGF-β signaling can also render the transition back

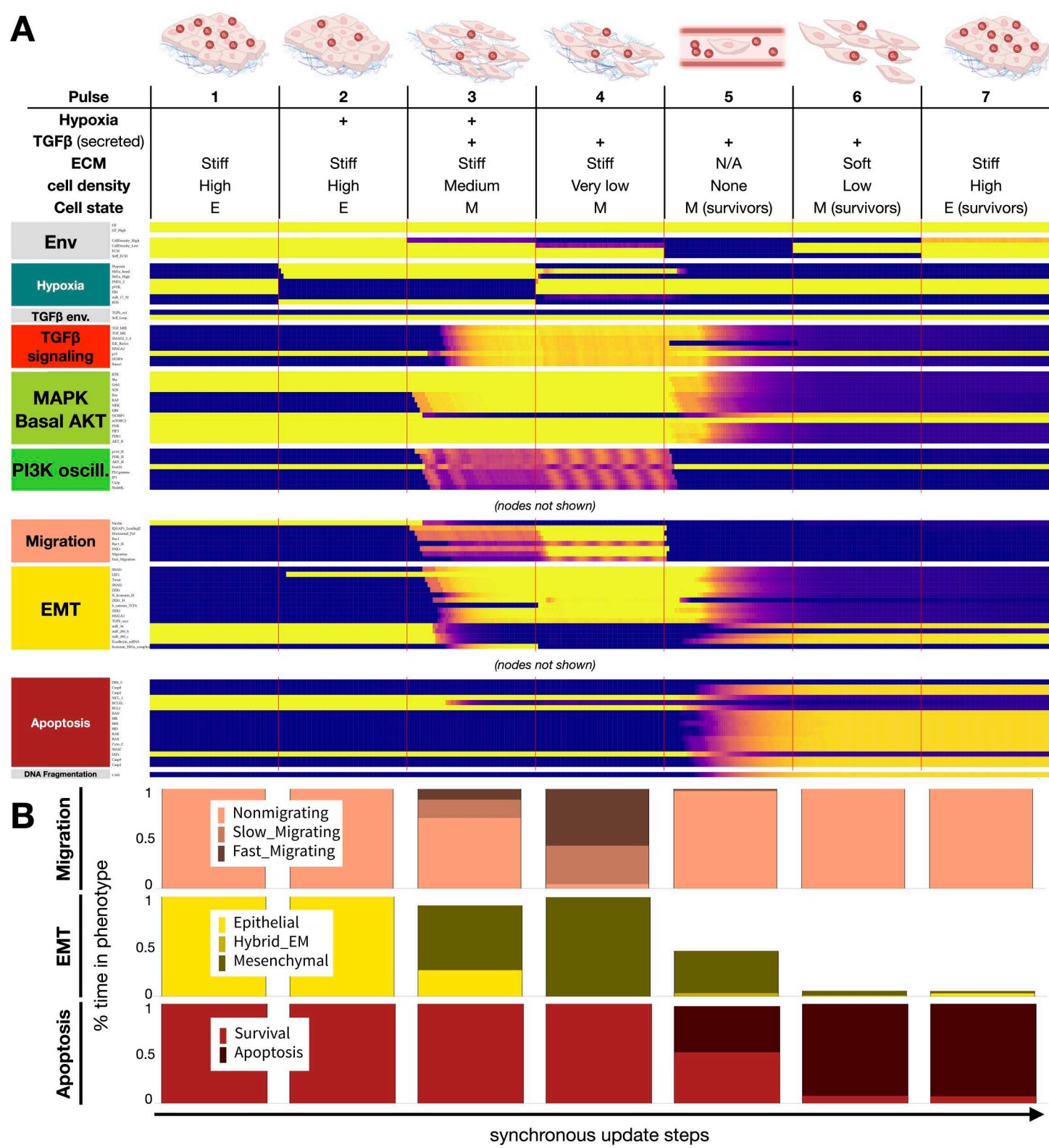

**Fig 6. As cells follow the metastatic cascade, hypoxia-induced EMT protects cells in circulation, reversed at the secondary site by high cell density.** A) Dynamics of relevant regulatory molecule expression in an ensemble of 1000 quiescent epithelial cells exposed to a sequence of

microenvironments along the metastatic cascade (100 update steps per pulse). *Pulse 1:* normoxia, stiff ECM, high density; *Pulse 2:* hypoxia, stiff ECM, high density; *Pulse 3:* hypoxia, stiff ECM, medium density; *Pulse 4* (invasion): normoxia, stiff ECM, very low density; *Pulse 5* (intravasation): normoxia, no ECM, no neighbors; *Pulse 6* (extravasation): normoxia, soft ECM, low density; *Pulse 7* (MET): normoxia, stiff ECM, high density; *X-axis:* update steps; *y-axis:* nodes organized by regulatory module; *yellow/blue:* ON/OFF; *black/white labels:* relevant phenotypes; *update:* synchronous; *autocrine TGF-β:* 5% TGFb_secr knockdown; *full time-course:* S8 Fig. B) Average fraction of time an ensemble of 1000 cells spend in *top:* non-migrating (*light pink*), slow-migrating (*dark pink*) and fast-migrating (*brow-pink*) states; *middle:* epithelial (*yellow*), hybrid E/M (*dark yellow*) and mesenchymal (*mustard*) states; *bottom:* survival (*red*) vs. in apoptotic (*dark red*) states for each pulse along the metastatic cascade in panel **A**. *Image credits:* metastatic cascade adapted from https://commons.wikimedia.org/wiki/File:Contribution_of_EMT_to_cancer_progression.jpg.

onto a soft ECM dangerous, as a weaker mesenchymal program is less apt at warding off the apoptotic effects TGF-β has on epithelial cells on soft ECM [86].

A particular strength of our model is its ability to reproduce a wide array of experimental results in wild-type, as well as mutant cells. Namely, the model matched the direction and significance of molecular or phenotypic changes in 138 of 150 independently conducted experimental assays from 13 primary papers focused on hypoxia and EMT (S1 Table), and 7 of its 12 failures were due to contradictory experiments in our list. This validation lends weight to our predictions, summarized below:

a) We predict that the loss of TGF-β signaling can weaken hypoxia-mediated cell cycle arrest (Fig 3D), and slow down hypoxia-induced EMT in cells with weak mitogenic signaling (Fig 3B-3C).

b) We predict that soft ECMs drastically raise the hypoxia threshold of EMT, while they impact TGF-β more mildly. On very soft ECMs unable to support stress fiber formation and migration, only a combination of strong hypoxia and near-saturating TGF-β can induce EMT (S5B Fig).

c) We predict that breaking the cell cycle inhibitory effects of VHL loss requires combined hyper-activation of Myc *and* Cyclin D (S7 Fig; less potent 2-gene combinations involve either p21 loss or Cyclin D hyper-activation). In contrast, upstream mutations that boost PI3K or MAPK signaling (alone or in combination with Myc) cannot overcome cell cycle arrest mediated by Hif-1α and/or TGF-β.

d) We predict that epithelial cells on a soft ECM exposed to TGF-β are protected from apoptosis under hypoxic conditions (Fig 5B).

e) Finally, we predict that circulating mesenchymal cells are highly sensitive to intermittent weakening of autocrine TGF-β signaling, and thus only a fraction of them survives anoikis (S9 and S10 Figs). This could be tested with cell suspension experiments on cells that first undergo EMT, treated with increasing doses of TGF-β or TGF-β RI inhibitors.

Our use of a Boolean modeling framework, despite its advantages in allowing us to build a 165-node regulatory network with 763 links, presents two technical limitations of particular interest here (also discussed in [86]). *First,* modeling autocrine signaling in a Boolean framework means that a positive feedback loop can lock in between a signaling pathway and the transcription/secretion of the signal itself, such as TGF-β. This loop implicitly assumes that a single mesenchymal cell produces sufficient levels of bio-available TGF-β to maintain saturating signaling on its own receptors, a likely problematic assumption. To get around the resulting artifacts, we intentionally disrupted autocrine signaling using a 5% reduction in secreted TGF-β in most simulations. This confers some reversibility to EMT, permitting a fluid coexistence of mesenchymal and hybrid E/M cells, the latter of which can divide. Yet, it is possible that *in vitro* the slow division of mostly mesenchymal cells is not due to temporary loss of TGF-β signaling (as modeled), but to the fact that even saturating levels of TGF-β do not fully block cell cycle entry in mesenchymal cells.

A second technical limitation, detailed in [86], relates to our Boolean approximation of ECM stiffness and cell density. There are three qualitatively distinct regimes for these inputs: no ECM/ soft ECM not capable of supporting stress fibers

and growth/ stiff ECM that does. Similarly for density: no neighbors capable of making junctions/ some neighbors but also some space/ full confluency with no room to expand. Our Boolean model uses two linked nodes for each input, generating a 0/ low/ high state. This setup results in distinct signals in the 0 to low vs. the low to high regimes without overlap, and more rigid model responses near these boundaries than expected *in vitro.*

A different class of model limitations relates to the signaling biology we choose to include or ignore. *First,* the five experiments our models failed to match that did not contradict other data highlighted that at present our model is missing the molecular mechanism responsible for Hif-1α-mediated G2 arrest (our model only arrests in G1), as well as hypoxia-induced apoptosis (not modeled by choice). *Second*, we restricted our focus to the *acute* hypoxic response governed primarily by Hif-1α stabilization. Chronic hypoxia leads to a different set of responses our model does not cover, as they are induced by Hif-2α stabilization [136]. Despite some conflicting data, it is generally accepted that the third HIF subunit, Hif-3α, prevents Hif-1α transcriptional activity through sequestration of Hif-1β in a cell-type-specific manner [137,138]. Expansion of the model to include these Hif-α subunits is a promising future direction, required for modeling the temporal control of cell fates under hypoxia.

*Third,* Hif-1α is a metabolic regulator, yet we only included its effects on energy metabolism indirectly. The inclusion of a nutrient stimulus such as glucose, and the energy-sensing pathways that intersect with cell cycle entry modeled, in part, in [139] would allow for a more precise accounting of the hypoxic response. For example, apoptosis is differentially regulated based on nutrient loss in the presence or absence of Hif-1α, as Hif-1α prevents metabolic stress-induced apoptosis [95]. Glucose transporters GLUT1/3 are also upregulated in mesenchymal cells, demonstrating a shift towards glycolysis during EMT and suggesting a dependence of EMT on nutrient levels under hypoxia and/or normoxia [140]. A future expansion of our model could cover these responses, further mapping the environment combinations most conducive to EMT and MET.

*Fourth,* our model excludes several signaling pathways known to be differentially regulated under hypoxia, such as Wnt, Sonic Hedgehog (Shh), Notch, Interleukin-6, or TNF-α [141–148]. Thus, we operate under the implicit assumption that these signals are induced at equal levels in response to all stimuli tested within our model – even though their regulation may also be mechanosensitive. These signals are also implicated in mediating EMT, preventing apoptosis, and contributing to the spatial and temporal regulation of EMT in response to a changing TME [149]. Overall, the integration of these pathways may offer further contextual insight into cancer-associated behaviors increased by hypoxic tumor microenvironments.

The challenges of applying a cell-type agnostic model of hypoxia-induced EMT to specific carcinomas are at least three-fold. First, the most prevalent driver mutations and pathway alterations driving carcinoma vary substantially by tissue (e.g., VHL loss in ccRCC vs. BRAF^V600E mutations in melanoma). Moreover, our current model does not cover pathways that regulate genomic instability or immune evasion [150]. Second, there is substantial genetic diversity within tumors of the same type; not only among patients but also among cells from a single tumor [151,152]. Third, tumor growth is a cell population-level phenomenon, where cell-cell interactions go beyond contact inhibition or EMT-promoting signaling (modeled here), extending to signals from tumor-associated fibroblasts and immune cells (not modeled). Given these challenges, it is important to address: *i)* which, if any, tumor cell type do we expect our predictions to apply to, and *ii)* how can our model answer questions about specific cancers? *i)* Our predictions address mechano-sensitive responses such as EMT or proliferation under hypoxia (b,d,e) or the effects of specific mutations (a,c) in controlled conditions. Thus, we expect our predicted trends to hold *in vitro* in several carcinoma cell lines, with substantial variation in effect size and/or the environments the effects are strongest in (note that most mutation combinations only change a subset of model behaviors, if any; S7 Fig). *ii)* Given the challenges of predicting cancer cell behaviors *in vivo*, we expect our model to be of use in predicting the *range* of cell behaviors that mutations associated with a specific carcinoma can generate, in combination, in different microenvironments (e.g., S7 Fig for ccRCC), in response to chemotherapy/targeted therapy. These predicted responses can then be compared to single-cell RNA sequencing from tumors [153], analyzed with a focus on cell-behavior signatures rather than individual genes (e.g., EMT, cell cycle, apoptosis). This can help us pinpoint enriched

behaviors compared to healthy tissues and predict mutation combinations that co-occur in single cells to drive tissue behavior (testable with DNA sequencing).

Single-cell transcriptomics can also identify key tissue-specific cancer cell signaling pathways missing from our model, pointing to cell-cell interactions we most need to account for. As an example, our *in silico* metastatic cascade is a simplification of the highly complex tumor and blood microenvironments experienced by circulating tumor cells. Extensive research has yielded both increasing and decreasing metastatic potential through interactions with neutrophils, cancer-associated fibroblasts (CAFs), macrophages, platelets, and other cell types, all of which are missing from our single-cell modeling approach [154–157]. Critically, interactions with platelets can upregulate TGF-β signaling and EMT, which help prevent cell death during the circulating phase [158].

The model boundaries highlighted above stem, in part, from our single-cell approach. We find this single cell focus valuable for mapping the rules that govern cell behavior in different microenvironments, a critical step towards building multi-scale models of cell communities and tissue architectures needed to approach the complexity of tumor formation, tumor evolution, and metastatic disease. Long-term, our goal is two-fold. On the *single-cell* modeling side, we plan to integrate the model from this study with our Boolean model of mitochondrial dysfunction-associated senescence, which shares this model's growth signaling, cell cycle, and apoptosis modules [139]. We will follow this with a detailed model of damage-induced cell cycle arrest mechanisms and a model of deep senescence, which can reverse mitochondrial dysfunction [159], but it is at least partially blocked in cells that underwent EMT [160,161]. The resulting single-cell model would be able to address most cell-autonomous cancer hallmarks. On the *tissue* modeling side, the above single-cell model can serve as the engine inside agents of the multicellular, spatial model of epithelial homeostasis and carcinoma development.

## Computational methods

I. ***Boolean model building.*** To build our model we extended our previously published mechanosensitive EMT model [86] with a hypoxia-sensing signaling module. The model synthesizes experimental data from 540 papers into a 165-node Boolean network with 763 regulatory links. All links and regulatory functions are experimentally justified in S1 Text. Our model construction approach, the rationale for using synchronous update, and storing our model in *Dynamically Modular Model Specification* (.*dmms*) format are detailed in the *Supplementary Methods* of [139] and STAR Methods of [86]. All model files and other files required to run the model and reproduce all results are packaged into S1-S3 Files.

II. ***Model availability.*** Model files in *SBML* format (used by BioModels [162], *GinSim* [163], and *The Cell Collective* [164]), *dmms* format (used by our discrete-state modeling software *dynmod*), *BooleanNet* format (used by the BooleanNet Python library [165]), and editable network visualization in *graphml* format (read by yED [166]) are included in S1 File.

III. **Dynmod *Boolean modeling software.*** Boolean simulations and analysis were performed with the discrete-state modeling software *dynmod* [139], available on GitHub at https://github.com/Ravasz-Regan-Group/dynmod (our justification for using in-house software, along with instructions to install Haskell and compile *dynmod* are detailed in the *Suppl. Methods* of [139]). Briefly, our code can: **a)** automatically map each attractor to a combinatorial phenotype profile (e.g., quiescent, alive, MiDAS) based on user-defined signatures attached to regulatory switches; **b)** visualize and filter attractors of interest via their phenotype profiles, organizing them within a coordinate system of independent environmental input-combinations; **c)** set up simulations by specifying the initial environment and phenotypic state of the cell, rather than each node state; **d)** collect phenotype statistics on large ensembles of non-interacting cells (independent simulation runs) in non-saturating environments and/or non-saturating perturbations (e.g., 10% Hypoxia, 50% VHL inhibition); **e)** generate simulation-series that explore the model's behavior across a range of environmental inputs, combinations of 2–3 inputs, and/or the effects of increasing node knockdown/ hyper-activation (alone or in combination with changing environmental input levels; with/without additional

background mutations); and **f)** use metadata from *dmms* files to generate a formatted table with all biological documentation (S1 Text).To run simulations (including attractor sampling), *dynmod* parses user-generated experiment files (*.vex* format). The S1 File package includes two.*vex* our readers can use to reproduce all simulations used in our figures (*Hypoxia_EMT_Main_Fig. vex, Hypoxia_EMT_SM_Fig.vex*). Similarly, the S2 File package contains the isolated EMT module model (*EMT_Module_ Hypoxia.dmms*), the *EMT_Module.vex* file, and time courses that showcase the model attractors visualized in Fig 1B. Finally, S3 File contains the.*vex* file and script required for rerunning our validation experiments (see below). Additional information to run the simulations is available upon request from E.R.R.

IV. ***Attractor detection with* dynmod.** *Dynmod* uses synchronous update to find stable phenotypes and/or oscillations (attractors) via a stochastic sampling procedure [167] detailed in [86,168–170]. Briefly, we typically find attractors by running noisy time courses of length $T = 25$ with noise $p = 0.02$ from $N = 100$ different random initial conditions for each unique combination of environmental inputs ($N_{total} = 100*2^7 = 128,000$ random initial conditions). For each observed state along these time courses, the synchronous attractor basin is determined, recorded, and exported as a.*csv* file (*Greene_Hypoxia_Model_attractors_a100_25_2.0e-2.csv* in the S1 File package). To re-sample the model's attractors, edit the *Sampling{…}* command block near the top of *Hypoxia_EMT_Main_Fig.vex* to switch from the currently active *Read:* directive to the commented-out *Sample:* line.

V. ***Running simulations with* dynmod.** Precise use of each command is described in *Hypoxia_EMT_Main_Fig.vex* and *Hypoxia_EMT_SM_Fig.vex* (in S1 File), which are executed by *dynmod* using the **-e** command-line tag:

dynmod Hypoxia_EMT_Model.dmms -e Hypoxia_EMT_Main_Fig.vex

dynmod Hypoxia_EMT_Model.dmms -e Hypoxia_EMT_SM_Fig.vex

These files include instructions to simulate and visualize: a) synchronous time courses from a subset of cell states in a given initial environment, exposed to reversible changes in a single environmental signal (e.g., Fig 2A); b) non-saturating environments where an input is stochastically tuned between 0 and 1 (e.g., Fig 4A); c) partial or full knock-down/hyper-activation of arbitrary sets of nodes (e.g., Fig 3; justification and limitations in [86]); d) combine these in an arbitrary sequence of perturbations and environments defined in distinct time windows (e.g., Fig 6); e) average activity of all nodes and/or all module phenotypes in an ensemble of independent cells (e.g., Fig 2B); f) bar charts of the activity of user-specified nodes and/or module phenotypes, averaged over an ensemble of independent runs and across each distinct time window of an experiment (e.g., S3 Fig); g) bar charts and heatmaps showing the model's behavior (module phenotype states) across a range of environments and/or node perturbations (e.g., S6 Fig).

VI.  **Model validation.** To test the model's ability to reproduce experimentally observed cell behaviors and/or molecular changes related to EMT in general and hypoxic responses in particular, we curated a list of 150 distinct observations from 13 relevant primary articles (S1 Table). These observations cover knockdown or overexpression of 19 different proteins or microRNAs. For each *in vitro* experiment, we created matching *in silico* experiments or experiment pairs that mimicked the extracellular environment and perturbations such as knockdowns or overexpression. This extensive validation protocol is included in *Hypoxia_EMT_Validation.vex* (in S3 File). Running

dynmod Hypoxia_EMT_Model.dmms -e Hypoxia_EMT_ Validation.vex

generates simulation data on a population of cells for each experimental condition. To automate the 229 comparisons, each *in vitro* observation in S1 Table is accompanied by information about the name and location of its matching *in silico* data (experiment names, relevant time windows, code pointing to the correct attractor file). The Python script *Hypoxia Model Validation Script.py* reads a.*csv* version of S1 Table, as well as the simulation data, to perform unpaired t-tests, list, save, and summarize the validation results as the percentage of observations our model can

reproduce (script and.*csv* in S3 File). Simulation results are considered a match for the experiment if either of the following conditions are met: *i)* there is a significant change in the same direction as the *in vitro* result, and the change is at least 5% of the average of the two values compared, or *ii)* no change, no significant change, or smaller than 5% difference, matching an *in vitro* result showing no significant change. The script also generates figures for each comparison (S12 Fig); the 17 failed comparisons identified by the script were manually marked with red/orange "*Mismatch*" labels (*red*: model does not match biological outcome; *orange*: different experiments give conflicting results; model matches the other outcome).

VII. **Metadata table, 2D network visualization, and model conversion.**

a) To generate a formatted table with all model metadata included in the.dmms file, run

dynmod Hypoxia_EMT_Model.dmms -w -s

This generates a text file with metadata warning (e.g., missing descriptions, citations, unrecognized node/link types), and a folder containing the LaTeX and BibTeX reference files used to render S1 Text. To generate the pdf, use LaTeX or https://www.overleaf.com.

b) To generate a.gml file read by the network visualization software yED run

dynmod Hypoxia_EMT_Model.dmms -g

The resulting visualization can be modified in yED to alter node coordinates and node colors. As long as none of the links or groupings are altered in the editing process, and the changes are saved in.gml format, running dynmod Hypoxia_EMT_Model.dmms -u Hypoxia_EMT_Model.gmlwill read the altered coordinates and colors, then generate an updated.dmms file (with a new name). **Fig 1** was generated in yED by manual post-professing of the auto-generated.gml file (i.e., removal of groups, altered label colors for better contrast, creation of sub-networks).

c) To generate a.BooleanNet version of the model, run

dynmod Hypoxia_EMT_Model.dmms -t

d) To generate the.sbml version, use *bioLQM* (http://colomoto.org/biolqm/) with the command:java -jar bioLQM-0.8-SNAPSHOT.jar Hypoxia_EMT_Model_Fine.booleannet Hypoxia_EMT_Model.sbml

VIII. **Previously introduced regulatory modules.** Detailed descriptions of *Growth factor signaling*, *Replication origin licensing*, *Restriction switch*, *Mitotic phase switch*, *Apoptotic switch*, *Cell cycle processes, Adhesion, Contact Inhibition, EMT* and TGF-β signaling are detailed in [86].

## Supporting information

**S1 Fig. Hif-1α is required for proliferation under normoxia (full version of Fig 2A).**
(PDF)

**S2 Fig. Hif-1α hyper-activation blocks the cell cycle by repressing Myc.**
(PDF)

**S3 Fig. Hypoxia induces EMT independently of TFG-β.**
(PDF)

**S4 Fig. A stiff ECM aids hypoxia-induced EMT, compared to hybrid E/M under normoxia.**
(PDF)

**S5 Fig. ECM stiffness boosts both hypoxia- and TGF-β-induced EMT; high levels of both are required to drive EMT on very soft ECMs.**
(PDF)

**S6 Fig. VHL deficiency boosts bio mechanically induced EMT in normoxia as well as hypoxia at moderate ECM stiffness and medium-high density, but abolishes cell cycle entry.**
(PDF)

**S7 Fig. VHL deficiency-induced cell cycle arrest is broken by cooperative hyper-activation of Myc and Cyclin D.**
(PDF)

**S8 Fig. Hypoxia confers anoikis resistance and protection from TGF-β induced epithelial cell apoptosis on a soft ECM (full version of Fig 5).**
(PDF)

**S9 Fig. Saturating autocrine TGF-β signaling protects all mesenchymal cells from anoikis, but also blocks MET on a stiff ECM.**
(PDF)

**S10 Fig. Slight breaks in the autocrine TGF-β signaling loop are potent inducers of anoikis and apoptosis on soft ECM.**
(PDF)

**S11 Fig. A changing mix of model cell phenotypes as a function of mitogen, cell density, ECM stiffness and hypoxia allows us to track TME-driven EMT and MET during the metastatic cascade.**
(PDF)

**S12 Fig. Auto-generated validation figures (150 panels with statistical tests, marked for match/ mismatch with expediting data).**
(PDF)

**S1 Text. Description and experimental support for all nodes and links of the Boolean Regulatory network model in Fig 1C.**
(PDF)

**S1 Table. List of 150 published experimental assays used for model validation, marked to indicate success/failure of the model in replicating each assay.**
(XLSX)

**S1 File. Files read by *dynmod* to simulate the Boolean model's behaviors.** These include the model in.dmms format (as well as.BooleanNet and.SBML formats, namely *Hypoxia_EMT_Model.dmms*, *Hypoxia_EMT_Model.booleannet*, *Hypoxia_EMT_Model.sbml*), the "in silico protocol" files for reproducing all figures (*Hypoxia_EMT_Main_Fig.vex*, *Hypoxia_EMT_SM_Fig.vex*) and the model network in.graphml format (*Hypoxia_EMT_Model.graphml*).
(ZIP)

**S2 File. Files read by *dynmod* to simulate the isolated EMT module.** These include the EMT module model in.dmms format (*EMT_module_Hypoxia.dmms*) and an "in silico protocol" file for generating time courses to check the module's attractors (*EMT_Module.vex*).
(ZIP)

**S3 File. Files read by *dynmod* to rerun all model validation experiments (*Hypoxia_EMT_Validation.vex*), the Pyhton3 script and.csv file that performs statistical tests on the simulation results, summarizes them, and generates the figures in S12 Fig (*Hypoxia Model Validation Script.py, ST2 - Hypoxia model validation list.csv*).**
(ZIP)

## Acknowledgments

We thank Peter L. Regan for the ongoing improvement of the *dynmod* software package used to generate all modeling results.

## Author contributions

**Conceptualization:** Grant Greene, Ian Zonfa, Erzsébet Ravasz Regan.

**Data curation:** Grant Greene, Ian Zonfa, Erzsébet Ravasz Regan.

**Formal analysis:** Grant Greene, Ian Zonfa, Erzsébet Ravasz Regan.

**Funding acquisition:** Erzsébet Ravasz Regan.

**Investigation:** Grant Greene, Ian Zonfa, Erzsébet Ravasz Regan.

**Methodology:** Erzsébet Ravasz Regan.

**Project administration:** Erzsébet Ravasz Regan.

**Supervision:** Erzsébet Ravasz Regan.

**Validation:** Grant Greene, Erzsébet Ravasz Regan.

**Visualization:** Grant Greene, Erzsébet Ravasz Regan.

**Writing – original draft:** Grant Greene, Erzsébet Ravasz Regan.

**Writing – review & editing:** Grant Greene, Erzsébet Ravasz Regan.

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
