## [Decision Letter · Decision Letter 0]

4 Mar 2025

PCOMPBIOL-D-24-02209

A Boolean network model of hypoxia, mechanosensing and TGF-β signaling captures the role of phenotypic plasticity and mutations in tumor metastasis

PLOS Computational Biology

Dear Dr. Ravasz Regan,

Thank you for submitting your manuscript to PLOS Computational Biology. After careful consideration, we feel that it has merit but does not fully meet PLOS Computational Biology's publication criteria as it currently stands. Therefore, we invite you to submit a revised version of the manuscript that addresses the points raised during the review process.

Please submit your revised manuscript within 30 days May 04 2025 11:59PM. If you will need more time than this to complete your revisions, please reply to this message or contact the journal office at ploscompbiol@plos.org. Please include the following items when submitting your revised manuscript:

We look forward to receiving your revised manuscript.

Kind regards,

Sylvain Soliman

Academic Editor

PLOS Computational Biology

Stacey Finley

Section Editor

PLOS Computational Biology

**Additional Editor Comments :**

The reviewers point out several strengths of this paper and are supportive of acceptance. There are a few minor issues (Reviewer 3), which the authors are invited to address in a revised submission.

**Journal Requirements:**

1) We noticed that you used the phrase 'not shown' in the manuscript. We do not allow these references, as the PLOS data access policy requires that all data be either published with the manuscript or made available in a publicly accessible database. Please amend the supplementary material to include the referenced data or remove the references.

3) Some material included in your submission may be copyrighted. According to PLOSu2019s copyright policy, authors who use figures or other material (e.g., graphics, clipart, maps) from another author or copyright holder must demonstrate or obtain permission to publish this material under the Creative Commons Attribution 4.0 International (CC BY 4.0) License used by PLOS journals. Please closely review the details of PLOSu2019s copyright requirements here: PLOS Licenses and Copyright. If you need to request permissions from a copyright holder, you may use PLOS's Copyright Content Permission form.

Potential Copyright Issues:

i) Figures 6A, Supplementary Figures(9A, 10, and 11). Please confirm whether you drew the images / clip-art within the figure panels by hand. If you did not draw the images, please provide (a) a link to the source of the images or icons and their license / terms of use; or (b) written permission from the copyright holder to publish the images or icons under our CC BY 4.0 license. Alternatively, you may replace the images with open source alternatives. See these open source resources you may use to replace images / clip-art:

4)  Please amend your detailed Financial Disclosure statement. This is published with the article. It must therefore be completed in full sentences and contain the exact wording you wish to be published.

5) Please ensure that the funders and grant numbers match between the Financial Disclosure field and the Funding Information tab in your submission form. Note that the funders must be provided in the same order in both places as well. Currently, "the Faculty Development Funds, and Biochemistry and Molecular Biology Program" are missing from the Funding Information tab.

6)  Please include a completed 'Competing Interests' statement in the text box when submitting your production task, including any COIs declared by your co-authors, written in full sentences.

If you have no competing interests to declare, please state "The authors have declared that no competing interests exist". You may also provide an updated statement via email.

7) Please ensure that the Supplementary Files (1-4) are uploaded in a correct numerical order in the online submission form.

**Reviewers' comments:**

Reviewer's Responses to Questions

Reviewer #1: This manuscript proposes Boolean model that integrates hypoxic signaling with a mechanosensitive model of

EMT. The model is impressively comprehensive, it which includes the EMT-promoting crosstalk of mitogens and biomechanical signals, cell cycle control, and apoptosis. The model is validated by showing that it reproduces multiple phenotypes and processes, for example hypoxic protection from anoikis.

The model has high explanatory power, as well as predictive power. For example, the model yields experimentally testable predictions about the effect of loss of the Von Hippel Landau protein (VHL) on cancer hallmarks, with or without secondary oncogene activation.

This model is a key advance, integrating multiple cancer hallmarks. It can be further extended to cover DNA damage response and senescence, which would then cover all single-cell cancer hallmarks.

The methods and results are well-documented and the writing is clear.

In addition to its clear biological/medical implications, this manuscript is a very good example of successful modeling.

Reviewer #2: In this work, the authors built a Boolean network model that extends their previously published model to integrate a hypoxic response module and study the dynamics of signaling pathways associated with the epithelial-mesenchymal transition.

Their model successfully reproduces opposite outputs that depend on multiple signals and integrates the influence of biomechanical features of the environment, such as the stiffness of the extracellular matrix. The links and nodes in the Boolean network can be traced to experimental evidence in the literature, and the authors validated the results of their simulations by matching them to experimental data.

The article is well-structured and well-written. The introduction is thorough and provides the reader with enough context to understand the results, particularly the paradoxical hypoxia responses. Every claim in the results can be easily traced to a figure, and the computational methods and tools that were used are well-described and referenced. Additionally, the supplementary files include all the data needed to run the model and ensure reproducible results. The Discussion section, although somewhat lengthy, recapitulates the results, comments on their relationship with the hallmarks of cancer, and discusses the limitations of the model. Overall, I must admit that I really enjoyed reading and reviewing this article. It is a great example of the relevance of Boolean networks and their application in cancer systems biology, and I hope the authors continue expanding the use of this model.

Reviewer #3: The manuscript presents a single-cell Boolean network model that incorporates hypoxia signaling in a previously published mechanosensitive epithelial to mesenchymal transition (EMT) model. The authors aim to model the signaling crosstalk between environmental factors, such as extracellular matrix stiffness, cell density, hypoxia, and growth signals, that are central for tumor metastasis, and they also aim to model the effect of these factors on the cell cycle, cell death, and EMT.

The integration of these processes in a single model is a significant advancement; in the literature, these processes are mostly studied in isolation. The authors used their in-house simulation software dynmod and provided sufficient detail for the reproducibility of their work. The model underwent extensive validation and could reproduce key findings from the literature. The authors provide predictions for interactions, between environmental factors and mutations, that affect cancer hallmarks. Original experimental validations for the model and predictions were not included.

Overall, this manuscript is written well for the most part and its results provide significant insight. However, it has a few issues, which the authors should address so that it can be recommended for publication.

Major issues:

1. The authors include a detailed discussion about the limitations of the model; however, an extra discussion should be added about the applicability of the predictions of a general model to a specific type of cancer cell, which can have numerous cell-type specific interactions that are not captured by the general model.

2. In line 511, “apoptosis of a neighbor” is mentioned as an example of how an epithelial cell can gain the room necessary for EMT. However, on Figure 6, no apoptosis can be observed in the corresponding pulse, Pulse 3. This may lead to confusion, and the authors should clarify how cell density can decrease at this pulse.

Minor issues:

1. On page 8, the sentence: “Under moderate hypoxia, localization of Hif-1⍺ to the mitochondria prevents oxidative-stress induced apoptosis through…” seems to be unfinished.

2. Figures 2, 3, 4, 6: y axis is labelled “% time in phenotype”, but values are 0–1 and not in percentage. Figure 4B: values along the y axis are partially obstructed. On SM Fig. 5B-C the x axis label is missing.

3. In line 300, authors should introduce the abbreviation SAC for those unfamiliar.

4. On page 28, the line “We predict that the loss of TGF-β signaling can… slow down hypoxia-induced EMT in cells with weak mitogenic signaling (SM Fig. 3).” seems to incorrectly refer to SM Fig. 3 based on the line on page 15: “…TGF-β receptor I knockdown does not block hypoxia-induced EMT – though it slows it in mild hypoxia (Fig. 3B-C).”

5. Authors should proofread the manuscript for typos and omissions.

**Have the authors made all data and (if applicable) computational code underlying the findings in their manuscript fully available?**

Reviewer #1: Yes

Reviewer #2: Yes

Reviewer #3: Yes

PLOS authors have the option to publish the peer review history of their article (what does this mean? ). If published, this will include your full peer review and any attached files.

**Do you want your identity to be public for this peer review?** For information about this choice, including consent withdrawal, please see our Privacy Policy .

Reviewer #1: No

Reviewer #2: **Yes: ** Diana Garcia-Cortes

Reviewer #3: No

**Figure resubmission:**
---

## [Editor Report · Decision Letter 1]

26 Mar 2025

Dear Whitmore-Williams Assistant Professor Ravasz Regan,

We are pleased to inform you that your manuscript 'A Boolean network model of hypoxia, mechanosensing and TGF-β signaling captures the role of phenotypic plasticity and mutations in tumor metastasis' has been provisionally accepted for publication in PLOS Computational Biology.

Best regards,

Sylvain Soliman

Academic Editor

PLOS Computational Biology

Stacey Finley

Section Editor

PLOS Computational Biology

---

## [Editor Report · Acceptance letter]

PCOMPBIOL-D-24-02209R1

A Boolean network model of hypoxia, mechanosensing and TGF-β signaling captures the role of phenotypic plasticity and mutations in tumor metastasis

Dear Dr Ravasz Regan,

I am pleased to inform you that your manuscript has been formally accepted for publication in PLOS Computational Biology. Your manuscript is now with our production department and you will be notified of the publication date in due course.

With kind regards,

Anita Estes
